# What You Pretrain On Matters: Synthetic Task Distributions Determine Tabular Foundation Model Quality

**Mohamed Bouadi** [1]  **Nassim Bouarour** [1]  **Shivam Dubey** [1]  **Varun Kulkarni** [1]  **Aditya Tanna** [1]
**Vinay Kumar Sankarapu** [1]

## Abstract

Tabular foundation models acquire their inductive biases almost entirely from synthetic pretraining distributions, yet prior design remains poorly understood. Standard synthetic priors are too well-behaved: they omit the confounding, structured missingness, distributional shift, and spurious support-query correlations that characterize real tabular data. We introduce O'PRIOR, a compositional realism prior built around four coupled components: a hierarchical SCM meta-generator spanning diverse functional families; a modular realism engine covering heterogeneous marginals, missingness, and target transforms; an explicit stress module injecting confounding and support-query mismatch; and a curriculum-governed, leakage-safe generation protocol. Holding architecture, optimizer, and compute budget fixed, O'PRIOR yields consistent and substantial improvements across real tabular benchmarks. Ablations confirm that mechanism diversity, realism composition, and shift-aware stress each contribute independently, establishing synthetic prior construction as a first-order and largely overlooked determinant of tabular foundation model quality.

## 1. Introduction

Gradient-boosted trees remain the dominant baseline for supervised tabular prediction, a gap resistant to the deep learning advances that transformed vision and language (Grinsztajn et al., 2022; McElfresh et al., 2023). Tabular foundation models (TFMs) offer a structurally different approach: after pretraining on synthetic supervised tasks, the model conditions on a labeled support set and predicts query labels in a single forward pass – no dataset-specific

training, no hyperparameter tuning (Grinsztajn et al., 2025; Qu et al., 2025; Zhang et al., 2025) – now matching or exceeding well-tuned tree baselines (Erickson et al., 2025a). This raises an immediate question: *what determines the quality of a pretrained tabular model?*

Unlike language or vision, TFMs are not pretrained on naturally occurring data. Their inductive biases are induced almost entirely by synthetic tasks sampled from a hand-designed generative prior, not merely a data source, but the mechanism through which the model learns what tabular prediction looks like. This makes prior design a first-order scientific question, yet it has received almost no study.

**Prior design is underspecified.** Standard synthetic priors, from the MLP-based SCMs of TABPFN (Hollmann et al., 2023) to the tree-augmented priors of TABFORESTPFN (den Breejen et al., 2024) and the richer transforms of TABICL (Qu et al., 2025), generate worlds that are too well-behaved: smooth marginals, no missingness, no spurious correlations. When TABPFN-V2 enriched its prior alongside architectural changes (Hollmann et al., 2025), gains could not be attributed to either factor independently. The broader generative modeling literature, GANs (Xu et al., 2019), diffusion (Kotelnikov et al., 2023), autoregressive generators (Borisov et al., 2023), targets fidelity to a fixed dataset rather than diversity of supervised learning problems, and is not directly applicable.

This paper treats prior design as the primary scientific object and introduce O'PRIOR, a compositional realism prior combining a hierarchical SCM meta-generator, a modular realism engine, and a distribution-shift and shortcut-stress module, all governed by a curriculum-controlled leakage-safe protocol. Holding architecture, optimizer, and compute fixed, O'PRIOR yields consistent and substantial improvements over simpler baselines, with ablations confirming that mechanism diversity, realism composition, and shift-aware stress each contribute independently.

## 2. O'PRIOR: Compositional realism priors for tabular pretraining

Real tabular datasets are not clean samples from smooth input-output functions. They are shaped by latent causal

[1]Lexsi Labs - Mumbai, Paris, London. Correspondence to: Mohamed Bouadi <mohamed.bouadi@lexsi.ai>.

*Proceedings of the 2nd ICML Workshop on Foundation Models for Structured Data*, Seoul, South Korea. 2026. Copyright 2026 by the author(s).

structure, heterogeneous feature distributions, structured missingness, measurement artifacts, and distributional shifts between training and test populations. Standard synthetic priors address functional diversity but systematically neglect the rest. O'PRIOR closes this gap through a sequential pipeline of four components, each motivated by what the previous stage leaves unresolved.

**Leakage-safe contract.** A TFM observes a labeled support set $\{(\mathbf{x}_t, y_t)\}_{t=1}^n$ and predicts labels for query rows $t > n$, unseen at inference time. Any preprocessing statistic computed over query rows constitutes information leakage. Throughout O'PRIOR, all fitted statistics are $\mathcal{F}_n$-measurable, computed exclusively from support rows, and query rows are transformed only by:

$$\tilde{x}_{tj} = h_j(\{x_{sj}\}_{s \leq n}, x_{tj}), \quad t > n. \tag{1}$$

We state this contract once; all components respect it without re-statement. Each synthetic task is a tuple $\mathcal{D} = (\mathbf{X} \in \mathbb{R}^{T \times d}, \mathbf{y} \in \mathbb{R}^T, n)$, where rows $t \leq n$ form the support and rows $t > n$ the query set.

### 2.1. Structural causal model generator

The first stage establishes functional and causal diversity: without it, all downstream realism perturbations would apply to a single impoverished family of mechanisms. A prior $p(\theta)$ over mechanisms $\Theta$ induces a distribution over supervised tasks: $\theta \sim p(\theta)$ , $(\mathbf{X}^{(0)}, \mathbf{y}^{(0)}) \sim p_\theta(\mathbf{X}, \mathbf{y})$.

In O'PRIOR, mechanisms are structural causal models (SCMs) (Pearl, 2009; Peters et al., 2017): directed acyclic systems where latent causes, observed covariates, and the response are generated from parent variables and independent exogenous noise. Tasks are drawn in three nested levels: groups share a meta-hyperprior draw $\boldsymbol{\omega} \sim G$ over structural and noise hyperparameters; subgroups share a mechanism class $m \in \mathcal{M}$ and dimension $d$; individual tasks draw $(T, n, K)$ conditionally. Since many components of $\boldsymbol{\omega}$ are themselves random – parameters of sampling distributions are drawn before operational hyperparameters – the prior exhibits hyperprior variability: groups experience systematically different depths, sparsities, and noise scales without fixing a universal grid.

**SCM-based Models.** O'PRIOR draws from five base classes, covering the functional relationships found in real tabular data: (i) **MLPs**, first used by (Hollmann et al., 2023), with random Gaussian weights and layer-wise noise; (ii) **Decision Trees** extending XGBoost-only used by Qu et al. (2025) with Decision Tree (Breiman et al., 2017), Extra Trees (Geurts et al., 2006), Random Forest (Breiman, 2001), and directly sampled variants; (iii) **Convolutional** layers with 1D kernels for local feature dependencies (Zhang et al., 2025); (iv) **Gaussian Processes** with RBF kernels for smooth low-dimensional structure (Williams & Ras-

mussen, 2006); and (v) **Time-Lagged VAR** for temporal and autoregressive patterns (Teräsvirta et al., 2010).

**Hybrid SCM.** Each base class generates tasks through a single functional family, but real tabular datasets are rarely so uniform, a financial dataset may combine linear macroeconomic relationships with polynomial covariate interactions, while a clinical dataset mixes tree-structured treatment rules with GP-like biological variation. The *Hybrid SCM* composes multiple base classes within a single task: an ordered tuple $(\theta^{(1)}, \ldots, \theta^{(\ell)})$ defines a random DAG whose edges span different structural families, with node values aggregated via mean, weighted softmax, MLP, product, or max operators. Features and targets are selected using diversity-aware strategies – k-means, farthest-point selection, entropy-based (Kraskov et al., 2004), and graph community detection (Traag et al., 2019) – ensuring selected features span diverse representational regions rather than clustering around a single mechanism.

### 2.2. Modular realism engine

The SCM stage produces functionally diverse but idealized tasks: marginals are approximately Gaussian, missingness is absent, and no observational artifacts are present. Real tabular data differs from this in ways that directly affect what predictive signals a model can extract. The realism engine maps each raw SCM output to an observational table via:

$$(\mathbf{X}^{(0)}, \mathbf{y}^{(0)}) \longmapsto (\tilde{\mathbf{X}}, \tilde{\mathbf{y}}, \mathbf{M}, \boldsymbol{\tau}), \tag{2}$$

where $\mathbf{M} \in \{0, 1\}^{T \times d'}$ is a missingness mask and $\boldsymbol{\tau}$ stores per-column semantic type and imputation identity.

**Preprocessing and marginal morphing.** Real tabular pipelines never consume raw generative latents, practitioners routinely apply clipping, standardization, and quantile transforms before modeling, and the resulting tables exhibit feature marginals that deviate strongly from Gaussian: heavy tails in financial data, count-like distributions in clinical records, bounded variables in survey responses. Omitting this preprocessing-and-morphing step would create a systematic mismatch between pretraining tasks and what the model encounters at test time. O'PRIOR therefore first applies a support-only preprocessing block: moments $\mu_j, \sigma_j$ define clipping thresholds $\tau_j^{\pm} = \mu_j \pm \kappa \sigma_j$, support rows are winsorized, and optional quantile maps and marginal transforms are fitted on support rows and applied to both splits. On top of this normalized base, selected columns undergo distributional morphing – heavy-tailed (Student-t , Pareto), bounded (Beta, sigmoid), count-like (Poisson, negative-binomial), ordinalized, and heteroscedastic perturbations of the form $\tilde{x}_{tj} = x_{tj} + \sigma_j \exp(\alpha|z_{tj}|)\epsilon_{tj}$ – alongside engineered columns such as polynomial interactions, low-rank projections, and noisy duplicates. Together these close the gap between the smooth idealized distribu-

tions that SCMs naturally produce and the heterogeneous marginals the model will encounter in deployment.

**Structured missingness.** For each column $j$, a mechanism $R_j \in \{\mathrm{MCAR}, \mathrm{MAR}, \mathrm{MNAR}\}$ is sampled from $\boldsymbol{\omega}$. MCAR entries are missing with fixed rate $\pi_j$; MAR missingness depends on a driver column $k(j) \neq j$ via $\mathbb{P}(m_{tj} = 1|\mathbf{x}_t) = \sigma(\beta_{\mathrm{MAR}} z_{t,k(j)} + b_j)$; MNAR depends on the feature's own value via $\mathbb{P}(m_{tj} = 1|x_{tj}) = \sigma(\beta_{\mathrm{MNAR}} s_j z_{tj} + b_j)$. Missing entries are imputed using support-only statistics with the strategy stored as column metadata, allowing the model to condition on missingness type.

**Target transformations and subgroup structure.** Targets undergo support-only normalization $y\tilde{y}_t = (y_t - \bar{y})/s_y$, followed by optional skew transforms, heavy-tailed noise, mixture perturbations, or censoring-style transformations. For classification, a latent score is discretized via support quantiles with optional label noise. An optional categorical subgroup attribute $g_t \in \{1, \ldots, G\}$ models demographic or institutional variation through group-specific target intercepts and projections.

### 2.3. Distribution Shift and Shortcut Stress

The realism engine improves individual task quality but does not alter the relationship between support and query rows. In practice, these populations routinely differ, through confounding, covariate shift, temporal drift, or spurious predictors informative in training but unreliable at test time. For an in-context learner these are not edge cases; they are the primary deployment failure modes. This stage injects structured support-query mismatch through four independently mechanisms: (1) **Latent confounding** adds a shared latent factor $z_t$ to a random subset of columns and to the target, inducing feature–target dependence that is not a direct causal effect. (2) **Spurious support predictors** construct columns that are strongly predictive on support $(\tilde{x}_{tj} = \lambda \tilde{y}_t + \epsilon_{tj})$ but degraded or sign-flipped on queries $(\tilde{x}_{tj} = s_j \rho \lambda \tilde{y}_t + \epsilon_{tj})$, stress-testing shortcut reliance; (3) **Covariate shift** applies query-only affine perturbations scaled to support dispersion; (4) **Temporal drift** adds seasonal periodicity or changepoint-based regime shifts, covering non-stationarity patterns common in financial and clinical data. All shift strengths are drawn from $\boldsymbol{\omega}$ and controlled by task-level Bernoulli switches, enabling clean isolation of each mechanism in ablations.

### 2.4. Curriculum and Leakage-Safe Generation

The three stages above define a rich family of synthetic tasks, but applying maximal realism uniformly from the start of pretraining is not necessarily optimal. Early in training the model has not yet learned basic predictive structure; immediate exposure to heavy confounding and adversarial shortcuts may slow convergence or prevent the

acquisition of fundamental in-context learning behavior. O'PRIOR therefore stages realism difficulty via three presets $\mathcal{P} \in \{\mathrm{LOW}, \mathrm{MILD}, \mathrm{HARD}\}$, each inducing a distinct hyperprior $G_{\mathcal{P}}$ by restricting the support of missingness rates, tail severity, confounding strength, and spurious feature fractions. During pretraining, the task hyperprior interpolates between profiles:

$$G_s = (1 - \alpha(s)) G_{\mathcal{P}_0} + \alpha(s) G_{\mathcal{P}_1}, \tag{3}$$

$\alpha(s) \in [0, 1]$ follows a linear, cosine, or step schedule. Crucially, the curriculum operates entirely on the prior, architecture, optimizer, and compute remain fixed, making it an independently ablatable experimental axis. Generated tasks failing basic validity checks (near-constant columns, collapsed support classes, degenerate target variation) are rejected and resampled, with checks kept deliberately minimal to avoid over-filtering toward an unrealistically clean task distribution.

## 3. Experimental Evaluation

### 3.1. Experimental Protocol

**Experimental setup.** We fix the model architecture, optimizer, and training budget, and vary only the synthetic task distribution used for pretraining. We use NANOTABPFN (Pfefferle et al., 2025) as our base TFM, a lightweight, open-source reimplementation of TABPFN V2, because it has a reduced scale making the pretraining efficient, and was designed for evaluating TFMs with controlled conditions.

We train all models on $40,000$ synthetic datasets per prior, each of size $T \in [512, 1024]$ rows and $d \in [3, 50]$ features, with a batch size of $4$ tables per optimization step, $1,000$ steps per epoch, and $10$ epochs of training, without hyperparameter tuning, using the TFM-Playground protocol[1].

**Baselines.** We compare against three generators: (i) **TabPFN-v1**, the original SCM-based prior from Hollmann et al. (2023), which generates synthetic datasets using MLP-based SCMs, (ii) **TabICL-v1** (Qu et al., 2025) which extends the MLP-based SCM prior with XGBoost-based structural equations, and (iii) **TabICL-v2** (Qu et al., 2026) which augments TabICL-v1 with additional SCM diversity and a richer set of feature-level transforms.

**O'Prior ablation.** We construct nine variants by (i) isolating each generative component, (ii) combining them with a linear mild-to-hard realism curriculum, to identify the performance of each component. We use the following shorthand: **SM** (basic SCM families), **SH** (Hybrid SCM), **MR** (moderate realism), **SR** (strong realism), **SD** (distribution shift and shortcut stress).

**Benchmarks.** We evaluate on two benchmarks covering

---

[1]https://github.com/automl/TFM-Playground/

52 tabular classification tasks in total: (i) **TabArena-v0.1** (Erickson et al., 2025b), included in the TFM-Playground, comprising 21 datasets; (ii) **OpenML-CC18** (Bischl et al., 2017) from which we sample datasets satisfying: $d \in [2, 50]$ features and $N < 10{,}000$ samples, yielding 31 datasets. Details are provided in Appendix B.

**Metrics.** We report ROC-AUC, accuracy (ACC), and macro-averaged F1 score on test splits. For multi-class, ROC-AUC is computed using the one-vs-rest formulation. For each benchmark, results are averaged across its datasets.

### 3.2. Main Results

*Table 1.* Avg ROC-AUC, Accuracy, and F1-score for different compositions and variants of O'PRIOR.

| Variants | TabArena-v0.1 | | | OpenML-CC18 | | |
|---|---|---|---|---|---|---|
| | ROC-AUC | ACC | F1-score | ROC-AUC | ACC | F1-score |
| TabPFN | 0.5758 | 0.7257 | 0.3724 | 0.5189 | 0.5770 | 0.3105 |
| TabICL-v1 | 0.6361 | 0.7366 | 0.3842 | 0.5754 | 0.5883 | 0.3139 |
| TabICL-v2 | 0.7910 | 0.8026 | 0.5234 | 0.7240 | 0.6402 | 0.4442 |
| SM | 0.7881 | 0.8087 | 0.5555 | 0.7411 | 0.6641 | 0.4929 |
| SH | **0.8335** | 0.8349 | **0.6148** | 0.8228 | 0.7299 | 0.5789 |
| SM+SH | 0.8324 | 0.8298 | 0.5879 | **0.8313** | **0.7427** | **0.5906** |
| SM+MR | 0.8102 | 0.8152 | 0.5699 | 0.7967 | 0.7089 | 0.5473 |
| SM+SR | 0.8315 | 0.8331 | 0.6143 | 0.8128 | 0.7284 | 0.5848 |
| SM+SH+SR | 0.8295 | **0.8366** | 0.6138 | 0.8011 | 0.7045 | 0.5395 |
| SM+SD | 0.8012 | 0.8128 | 0.5400 | 0.7806 | 0.6969 | 0.5023 |
| SM+SH+SD | 0.8192 | 0.8250 | 0.5725 | 0.8064 | 0.7170 | 0.5271 |
| SM+SH+SD+ Curriculum | 0.8194 | 0.8212 | 0.5829 | 0.8245 | 0.7239 | 0.5693 |

**Ablation Analysis** Table 1 reports the controlled ablation study of the O'PRIOR pipeline. We observe that TABPFN performs weakest, TABICL-V1 improves modestly, and TABICL-V2 is stronger, especially on macro-F1 showing that prior design matters. Structural variants (i.e., SM and SH) are the largest single contributor outperforming the best baseline despite using no explicit realism or shifts. While hybrid SCM alone (SH) achieves the highest TabArena F1, combining it with basic SCMs (SM+SH) gives the strongest OpenML-CC18 ROC-AUC and ACC among the non-curriculum variants.

Adding realism to basic SCMs improves overall performance, favouring strong realism (SM+SR) over moderate ones (SM+MR) as the former is consistently better. This suggests heterogeneous marginals, more missingness and noise are more beneficial for learning. However, strong realism does not necessary improve performance of Hybrid SCM (SM+SH+SR) suggesting a nontrivial interaction between maximal realism and structural diversity. The shift module also improves over the basic SCMs (SM+SD), but with smaller gains than those of realism. While expected, this result reflects that shifts target support–query mismatch and shortcut failures rather than maximizing i.i.d. accuracy, making them a robustness-oriented component.

The curriculum variant does not dominate the best non-

curriculum variants (e.g., SH, SM+SH, SM+SR), but still outperforms the best baseline, improving over TABICL-V2 on every metric and benchmark. With fixed pretraining steps, while early curriculum training on milder profiles reduces exposure to the hardest realism and shifts, static strong priors can be more sample-efficient as they train from the final target distribution. Thus, curriculum is not a free improvement and may be more appropriate for longer pretraining runs or when optimization stability is important.

**Prior Quality via Structural Alignment.** We perform a schema-agnostic adaptation of TabStruct (Jiang et al., 2025), comparing O'PRIOR and TABICL-V2 against four real reference datasets to verify whether generated tables resemble real data. We report three scores: the *marginal score* measures one-dimensional feature distribution alignment via Wasserstein distance; the *correlation score* compares eigenvalue spectra of feature correlation matrices; and the *composite score $Q$* is their geometric mean, capturing both distributional and structural alignment jointly. Protocol details and plots are provided in Appendix A.1.

Table 2 shows that O'PRIOR achieves higher $Q$ and correlation scores, while TABICL-V2 obtains slightly higher marginal scores. The latter is an artifact of schema homogeneity: TABICL-V2's fixed dense format yields stable one-dimensional marginal estimates, whereas O'PRIOR's variable-schema generator spans a broader space of feature mechanisms, making marginal pooling inherently more heterogeneous. The structurally important signal is the correlation score: O'PRIOR better matches the eigenvalue decay and pairwise-correlation spread of real tables, while TABICL-V2 produces spectra closer to weakly dependent features, as shown in Figure 1.

To further assess whether the prior shapes internal model representations, we conduct a linear probing experiment extracting activations from three layers of NANOTABPFN pretrained on each prior. Models trained on O'PRIOR show monotonically increasing probing accuracy across layers on three out of four datasets, while TABICL-V2 exhibits flat or marginal improvement, suggesting that O'PRIOR induces richer and more structured internal representations. Full results are in Appendix A.2.

## 4. Conclusion

In this study, we showed that synthetic prior design is a central driver of TFMs quality. O'PRIOR improves transfer by combining diverse structural mechanisms with realism perturbations and shift-aware stress under a leakage-safe protocol. Our ablations indicate that these components, especially mechanism diversity and observational realism, substantially affect downstream performance. This positions prior construction as a key axis for future progress in TFMs.

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

# A. Aditional Experiments

## A.1. Additional Prior Quality Analysis

**Motivation.** The main results evaluate synthetic priors through the downstream performance of a trained tabular foundation model. This is the primary criterion for our study, but it does not directly measure whether the generated tables themselves resemble real tabular data. We therefore include a complementary structural-alignment analysis based on a schema-agnostic adaptation of TabStruct (Jiang et al., 2025). The goal is not to reproduce a particular dataset, but to test whether the prior generates tables with realistic marginal and dependency structure.

**Schema-agnostic adaptation.** TabStruct is originally designed for fixed-schema synthetic data evaluation. O'PRIOR, however, generates variable-schema supervised tasks with varying numbers of features, rows, and causal mechanisms. We therefore adapt the evaluation to operate on pooled statistics rather than matched columns. For each real reference dataset, we compare synthetic samples from Full O'PRIOR and TABICL-v2 along two axes: marginal alignment and structural correlation alignment. The main numerical results are reported in Table 2.

*Table 2.* TabStruct-adapted realism evaluation results. Scores are reported as mean $\pm$ standard deviation over 10 independent Monte Carlo iterations.

| Reference Data | Synthetic | Marginal | Correlation | $Q$ |
|---|---|---|---|---|
| Electricity (ID:151) | Full O'PRIOR | $0.979 \pm 0.003$ | $0.795 \pm 0.027$ | $0.882 \pm 0.015$ |
| Electricity (ID:151) | TABICL-v2 | $0.995 \pm 0.000$ | $0.760 \pm 0.025$ | $0.870 \pm 0.014$ |
| Diabetes (ID:37) | Full O'PRIOR | $0.974 \pm 0.003$ | $0.745 \pm 0.025$ | $0.852 \pm 0.015$ |
| Diabetes (ID:37) | TABICL-v2 | $0.988 \pm 0.002$ | $0.710 \pm 0.026$ | $0.838 \pm 0.015$ |
| Churn (ID:40701) | Full O'PRIOR | $0.971 \pm 0.003$ | $0.910 \pm 0.013$ | $0.940 \pm 0.007$ |
| Churn (ID:40701) | TABICL-v2 | $0.985 \pm 0.000$ | $0.887 \pm 0.021$ | $0.934 \pm 0.011$ |
| Home Credit | Full O'PRIOR | $0.957 \pm 0.002$ | $0.941 \pm 0.006$ | $0.949 \pm 0.004$ |
| Home Credit | TABICL-v2 | $0.968 \pm 0.001$ | $0.906 \pm 0.028$ | $0.937 \pm 0.015$ |

**Evaluation protocol.** We use four real reference datasets: Electricity (OpenML ID:151), Diabetes (OpenML ID:37), Churn (OpenML ID:40701), and Home Credit. For each synthetic source and each Monte Carlo iteration, we sample 50 synthetic tables. The marginal score pools feature values across sampled tables, caps the pool by reservoir sampling when necessary, quantile-normalizes values to $[0, 1]$, and compares the resulting distribution to the reference using Wasserstein distance:

$$S_{\mathrm{marginal}} = \exp(-W_1).$$

The correlation score extracts the eigenvalue spectrum of each synthetic table's feature correlation matrix, pools spectra across sampled tables, and compares this pooled spectrum to the reference spectrum:

$$S_{\mathrm{corr}} = \exp(-5W_1).$$

The composite score is the geometric mean

$$Q = \sqrt{S_{\mathrm{marginal}} S_{\mathrm{corr}}}.$$

We repeat the procedure over 10 independent Monte Carlo iterations and report mean $\pm$ standard deviation in Table 2.

**Interpretation of the table.** As shown in Table 2, O'PRIOR achieves higher composite quality $Q$ on all four reference datasets. The improvement is driven by correlation alignment: O'PRIOR obtains higher correlation scores on every reference, with especially large gains on Churn and Home Credit. These datasets have more features and richer inter-feature structure, which is precisely the regime where a compositional SCM prior should provide an advantage.

TABICL-v2 obtains higher marginal scores across references. We do not interpret this as stronger overall realism. Because TABICL-v2 uses a fixed dense schema, pooling 50 homogeneous synthetic tables produces very stable one-dimensional marginal estimates. O'PRIOR, by design, generates variable-schema tables with heterogeneous feature counts and mechanisms; pooled marginal statistics are therefore more diverse and can incur larger Wasserstein distances. This is a limitation of marginal-only evaluation for shape-agnostic priors. The composite score and correlation score better capture the structural objective of O'PRIOR: generating a broad distribution of realistic supervised tabular problems rather than matching a single fixed schema.

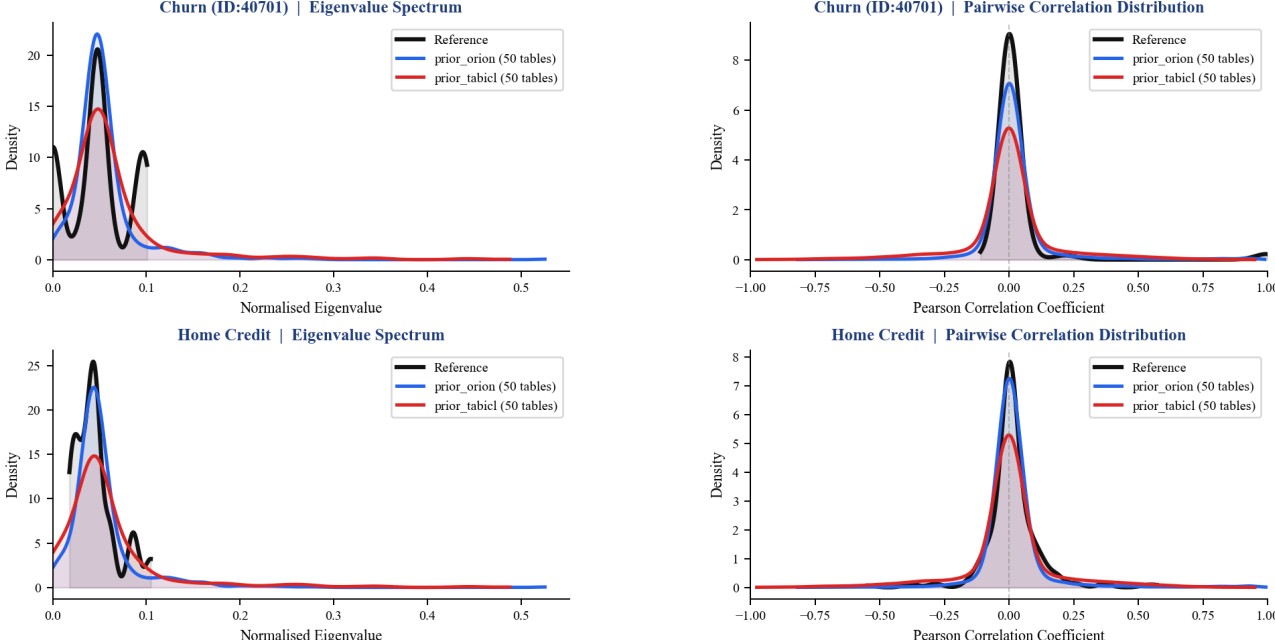

*Figure 1.* Structural alignment between real reference datasets and synthetic priors. Eigenvalue spectra and pairwise-correlation distributions are aggregated over 50 sampled tables per synthetic source. O'PRIOR better matches the eigenvalue decay and correlation spread of high-dimensional real datasets, while TABICL-v2 produces weaker inter-feature dependence.

**Structural visualizations.** Figure 1 compares the aggregated eigenvalue spectra and pairwise-correlation distributions for Churn and Home Credit. O'PRIOR more closely reproduces the characteristic structure of the real references: most eigenvalues concentrate near zero, while a controlled right tail captures dominant directions of feature dependence. The pairwise-correlation distributions show the same pattern. Real data and O'PRIOR both exhibit moderate spread around zero, consistent with sparse but nontrivial inter-feature dependencies. By contrast, TABICL-v2 is more concentrated near zero, indicating weaker dependence among generated features.

This analysis supports the mechanism behind the downstream gains observed in Section 3.2. O'PRIOR does not merely increase task diversity; it generates feature interaction patterns that are more structurally aligned with real tabular data. The results also show why evaluating variable-schema priors requires structural metrics in addition to marginal statistics: marginal alignment alone can favor fixed-format generators even when their dependency structure is less realistic.

### A.2. Prior Quality via TFM Internal Representations

While standard performance-based evaluation measures what a model achieves when trained on O'PRIOR, they still ignore the model internal representations. We thus conduct a linear probing experiment to evaluate whether NANOTABPFN internally captures complex data, or only learns simple patterns. We route four real-world datasets of varying domains and extract three layers from pre-trained models on O'PRIOR and TABICL-v2, and train a logistic regression on the target.

Table 3 reports the accuracy of the logistic regressors. O'PRIOR shows monotonically increasing probing accuracy across layers for Vehicle, Ionosphere, and Segment datasets. It outperforms the prior of TABICL-v2 which exhibits either a small or no increase. This suggests that models pre-trained on O'PRIOR are more robust to non-linear, high dimensional, multi-class datasets. Moreover, O'PRIOR shows clear above-random learning for these datasets while TABICL-v2 fails to do so for Vehicle and Segment. This confirms that our prior encourages informative feature learning.

## B. Benchmark Dataset Details

This appendix reports the datasets used in the experimental evaluation. The table includes datasets from the TabArena benchmark 4 and the OpenML-CC18 benchmark5. For each dataset, we provide the benchmark source, the resolved OpenML dataset identifier, dataset name, number of samples, number of features, number of classes, and the corresponding task type.

*Table 3.* Accuracy of probing logistic regressors across four real data and based on three layers.

| Reference Data | Synthetic | $L1$ | $L3$ | $L6$ | **Random** $L6$ |
|---|---|---|---|---|---|
| Vehicle (ID:54) | Full O'Prior | 0.33 | **0.48** | **0.67** | 0.39 |
| Vehicle (ID:54) | TabICL-v2 | **0.39** | 0.39 | 0.4 | 0.39 |
| Ionosphere (ID:59) | Full O'Prior | 0.653 | **0.807** | **0.884** | 0.692 |
| Ionosphere (ID:59) | TabICL-v2 | **0.73** | 0.769 | 0.73 | 0.692 |
| Segment (ID:36) | Full O'Prior | **0.32** | **0.68** | **0.83** | 0.55 |
| Segment (ID:36) | TabICL-v2 | 0.27 | 0.56 | 0.53 | 0.55 |
| Credit-g (ID:31) | Full O'Prior | 0.64 | 0.61 | 0.63 | 0.64 |
| Credit-g (ID:31) | TabICL-v2 | **0.66** | **0.64** | **0.64** | 0.64 |

*Table 4.* Summary of the selected TabArena-v0.1 classification datasets used in the evaluation.

| Dataset ID | Dataset name | Samples | Features | Classes | Task type |
|---|---|---|---|---|---|
| 46941 | maternal_health_risk | 1,014 | 7 | 3 | multiclass classification |
| 46905 | Amazon_employee_access | 32,769 | 10 | 2 | binary classification |
| 46906 | anneal | 898 | 39 | 5 | multiclass classification |
| 46952 | qsar-biodeg | 1,054 | 42 | 2 | binary classification |
| 46908 | APSFailure | 76,000 | 171 | 2 | binary classification |
| 46910 | bank-marketing | 45,211 | 14 | 2 | binary classification |
| 46911 | Bank_Customer_Churn | 10,000 | 11 | 2 | binary classification |
| 46912 | Bioresponse | 3,751 | 1,777 | 2 | binary classification |
| 46913 | blood-transfusion-service-center | 748 | 5 | 2 | binary classification |
| 46915 | churn | 5,000 | 20 | 2 | binary classification |
| 46916 | coil2000_insurance_policies | 9,822 | 86 | 2 | binary classification |
| 46963 | website_phishing | 1,353 | 10 | 3 | multiclass classification |
| 46918 | credit-g | 1,000 | 21 | 2 | binary classification |
| 46919 | credit_card_clients_default | 30,000 | 24 | 2 | binary classification |
| 46920 | customer_satisfaction_in_airline | 129,880 | 22 | 2 | binary classification |
| 46921 | diabetes | 768 | 9 | 2 | binary classification |
| 46922 | Diabetes130US | 71,518 | 48 | 2 | binary classification |
| 46980 | MIC | 1,699 | 112 | 8 | multiclass classification |
| 46924 | E-CommereShippingData | 10,999 | 11 | 2 | binary classification |
| 46927 | Fitness_Club | 1,500 | 7 | 2 | binary classification |
| 46938 | Is-this-a-good-customer | 1,723 | 14 | 2 | binary classification |

*Table 5.* Summary of OpenML-CC18 datasets used in the evaluation.

| Dataset ID | Dataset name | Samples | Features | Classes | Task type |
|---|---|---|---|---|---|
| 11 | balance-scale | 625 | 5 | 3 | multiclass classification |
| 15 | breast-w | 699 | 10 | 2 | binary classification |
| 18 | mfeat-morphological | 2,000 | 7 | 10 | multiclass classification |
| 23 | cmc | 1,473 | 10 | 3 | multiclass classification |
| 29 | credit-approval | 690 | 16 | 2 | binary classification |
| 31 | credit-g | 1,000 | 21 | 2 | binary classification |
| 37 | diabetes | 768 | 9 | 2 | binary classification |
| 50 | tic-tac-toe | 958 | 10 | 2 | binary classification |
| 54 | vehicle | 846 | 19 | 4 | multiclass classification |
| 188 | eucalyptus | 736 | 20 | 5 | multiclass classification |

| Dataset ID | Dataset name | Samples | Features | Classes | Task type |
|---|---|---|---|---|---|
| 307 | vowel | 990 | 13 | 11 | multiclass classification |
| 469 | analcatdata_dmft | 797 | 5 | 6 | multiclass classification |
| 1063 | kc2 | 522 | 22 | 2 | binary classification |
| 1067 | kc1 | 2,109 | 22 | 2 | binary classification |
| 1068 | pc1 | 1,109 | 22 | 2 | binary classification |
| 1462 | banknote-authentication | 1,372 | 5 | 2 | binary classification |
| 1464 | blood-transfusion-service-center | 748 | 5 | 2 | binary classification |
| 1480 | ilpd | 583 | 11 | 2 | binary classification |
| 1489 | phoneme | 5,404 | 6 | 2 | binary classification |
| 1497 | wall-robot-navigation | 5,456 | 25 | 4 | multiclass classification |
| 23381 | dresses-sales | 500 | 13 | 2 | binary classification |
| 40701 | churn | 5,000 | 21 | 2 | binary classification |
| 40975 | car | 1,728 | 7 | 4 | multiclass classification |
| 40982 | steel-plates-fault | 1,941 | 28 | 7 | multiclass classification |
| 40983 | wilt | 4,839 | 6 | 2 | binary classification |
| 40984 | segment | 2,310 | 20 | 7 | multiclass classification |
| 40994 | climate-model-simulation-crashes | 540 | 21 | 2 | binary classification |

