# OpenReview forum: "What You Pretrain On Matters: Synthetic Task Distributions Determine Tabular Foundation Model Quality"
_ICML.cc/2026/Workshop/FMSD — FMSD @ ICML 2026 Poster_

### Official Review · Reviewer_ejhV · 2026-05-13
**Advancing Tabular Foundation Models Through Better Synthetic Priors**

**Rating:** 6
**Confidence:** 4

**Review:**

## Summary
This paper studies the role of synthetic task distributions in pretraining tabular foundation models and proposes O’PRIOR, a compositional prior framework combining hybrid SCM generation, realistic marginals and missingness, distribution-shift stress, and curriculum-based difficulty scaling. Under fixed architecture and compute settings, O’PRIOR consistently improves downstream performance over strong prior baselines on tabular classification benchmarks.

## Strengths
- The paper addresses an important and relatively underexplored aspect of TFMs: prior design.
- The compositional prior framework is well motivated and includes several practically meaningful components such as realistic missingness and support-query shift stress.
- The hybrid SCM design substantially increases mechanism diversity compared to existing priors.
- Experimental design is careful, with architecture and compute held fixed to isolate prior effects.
- The ablations are informative and generally support the paper’s main claims.
- Structural-alignment and probing analyses provide additional evidence beyond downstream accuracy.

## Weaknesses
- Some components remain under-specified, especially the Hybrid SCM generation process and hyperprior choices.
- The evaluation focuses only on classification tasks and lacks regression experiments.
- Statistical significance and per-dataset variance are not reported.
- Robustness claims would be stronger with dedicated shift-evaluation benchmarks rather than only i.i.d. downstream testing.
- Handling of categorical and mixed-type features is not discussed in sufficient detail.

## Overall Assessment
I found this paper novel, well motivated, and practically meaningful. The main contribution is not a new architecture, but a convincing demonstration that synthetic prior design is a major factor in TFM quality. The proposed O’PRIOR framework introduces several thoughtful realism and robustness mechanisms, and the empirical improvements are supported by strong ablations and representation analyses. While some reproducibility and evaluation details could be strengthened further, I believe this is a valuable contribution that can influence future research on tabular foundation model pretraining.

---

### Official Review · Reviewer_Hr3p · 2026-05-21
**Review for What You Pretrain On Matters: Synthetic Task Distributions Determine Tabular Foundation Model Quality**

**Rating:** 7
**Confidence:** 4

**Review:**

## Summary
The paper focuses on an important topic in TFM, i.e., the synthetic data which powers them and forms the inductive biases. They propose
O’PRIOR, a compositional realism prior built around four coupled components: a hierarchical SCM meta-generator spanning diverse functional
families; a modular realism engine covering heterogeneous marginals, missingness, and target transforms; an explicit stress module injecting
confounding and support-query mismatch; and a curriculum-governed, leakage-safe generation protocol.

## Strengths
- The authors target an important problem in TFM which is often overlooked in the data and is the secret sauce of the synthetic data generation and aim to answer the question how TFMs can perform so strongly trained on only synthetic rather than LLM and CV models which are trained on real data.
- Good overview of TFM literature
- Nice diverse set of SCM-based priors ranging from MLP to tree-based to GP-based.
- Detailed ablation study to test the effect of each component.
- Good evaluation on state-of-the-art tabular dataset TabArena.

## Areas for Improvement
- Missing relevant reference to Zhang et al., "Mitra: Mixed Synthetic Priors for Enhancing Tabular Foundation Models", NeurIPS, 2025, which studies the effect of the priors and mixes the SCM with tree-based priors.
- It's true that the synthetic data is missing real-world challenges but it would be good for the authors to give more motivations on practical failure modes of TFMs on real-world data.
- ELO or winrate based metrics may be more meaningful to compare and add to Table 1 than the F1 score.

## Justification for Score
Overall, I think this paper is very well-suited for the workshop and highlights the important of the synthetic data generation on TFMs. It provides a rigorous experimental analysis of a mixture of SCM-based priors.

---

### Official Review · Reviewer_nin8 · 2026-05-22
**Review: What You Pretrain On Matters**

**Rating:** 5
**Confidence:** 4

**Review:**

## Summary
The paper argues that synthetic pretraining distributions are a primary determinant of tabular foundation model quality, and proposes O’PRIOR, a compositional realism prior for tabular pretraining. O’PRIOR combines hierarchical SCM generation, heterogeneous marginal/missingness/target transformations, distribution-shift and shortcut stressors, and a leakage-safe curriculum. The authors evaluate the prior while holding the model architecture, optimizer, and compute fixed, using NanoTabPFN and comparing against TabPFN and TabICL-style priors. The results show gains over the baselines on TabArena-v0.1 and OpenML-CC18, with ablations suggesting that structural diversity and realism components are especially important.

## Strengths
- The paper attempts to isolate structural diversity, realism, distribution shift, and curriculum effects, which is relevant to the workshop and the general community.
- The studied prior families cover several realistic tabular phenomena such as missingness.

## Areas for Improvement
- The novelty is limited. The central claim that the synthetic prior determines the behavior of a tabular foundation model is well known and has always been part of the design of new TFMs.
- The paper does not clearly distinguish which components are original versus inherited from prior tabular synthetic-data generation, O’PRIOR appears to combine many known ingredients for the prior like SCMs, tree/MLP/GP generators, marginal transformations, missingness, confounding, or covariate shift.
- The conclusions have limited practical use. Because the method is a large collection of heuristics, the study does not establish a scientific insight beyond “richer priors can help.” The claimed contribution would be stronger if the paper identified principled prior-design rules or provided empirical evidence that a specific (new) mechanism is essential.
- Sometimes non-standard terms are used without explaining them, e.g. “spurious support-query correlations”
- The utilized datasets are not explained further. Why are exactly those TabArena datasets used?
- Results per dataset not shown, where do the improvements really come from? Table 1 indicates huge improvements even over TabICLv2, yet it is unclear where these improvements stem from. What are the datasets where improvements are made and how large are the improvements on the individual datasets?
- It is unclear to me what the “leakage-safe contract” is, what problem it aims to solve, and what exactly existing TFMs miss in that regard. I don’t see where exactly information leakage that matters for pretraining would happen with the common design.
- The structural causal model generator seems to merely combine existing priors and adds some new ideas like time-lagged VAR that are not further explained or tested
- The Table 1 caption could be more expressive explaining what the row labels mean. Also, averaging over raw metrics is not ideal for aggregation since datasets have different irreducible error rates
- SH seems to deliver the largest gains. Given the limited experimental setup using NanoTabPFN and 40,000 sampled datasets per prior, I assume that that the results may be rather due to the generated datasets being a more efficient representation rather than adding anything new. Since the hybrid SCM just combines existing priors into one, the generated datasets contain patterns of multiple priors. So the convergence of the training can naturally be assumed to be faster. Without convergence trajectories for all models it remains unclear whether hybrid SCMs truly add value beyond faster convergence. To less extent, this applies to all the experiments.
- No accounting for uncertainty in the results. Without error bars, it is hard to judge the true extent and robustness of the results
- NanoTabPFN may not reflect how a more advanced architecture would behave under the studied prior.

## Justification of Score
The central claim of the paper is to show that that synthetic prior design is a central driver of TFMs quality. However, this is already known since the introduction of TFMs and is a known central component of the whole TFM modeling framework. The paper mixes known existing prior mechanisms with new ones, however I find it hard to distinguish what exactly is new and what is taken from prior work. Moreover, it remains unclear which prior improvements introduced in the paper have led to new improvements. At this stage I find it hard to infer actionable insights from the paper, which is why I tend towards rejection.